# Persistent Symptoms and IFN-γ-Mediated Pathways after COVID-19

**DOI:** 10.3390/jpm13071055

**Published:** 2023-06-27

**Authors:** Talia Piater, Mario Gietl, Stefanie Hofer, Johanna M. Gostner, Sabina Sahanic, Ivan Tancevski, Thomas Sonnweber, Alex Pizzini, Alexander Egger, Harald Schennach, Judith Loeffler-Ragg, Guenter Weiss, Katharina Kurz

**Affiliations:** 1Department of Internal Medicine II, Medical University of Innsbruck, Anichstrasse 35, 6020 Innsbruck, Austria; talia.piater@student.i-med.ac.at (T.P.); mario.gietl@student.i-med.ac.at (M.G.); sabina.sahanic@tirol-kliniken.at (S.S.); ivan.tancevski@tirol-kliniken.at (I.T.); thomas.sonnweber@tirol-kliniken.at (T.S.); alex.pizzini@i-med.ac.at (A.P.); judith.loeffler@i-med.ac.at (J.L.-R.); guenter.weiss@i-med.ac.at (G.W.); 2Institute of Medical Biochemistry, Biocenter, Medical University of Innsbruck, Innrain 80, 6020 Innsbruck, Austria; st.hofer@i-med.ac.at (S.H.); johanna.gostner@i-med.ac.at (J.M.G.); 3Central Institute for Medical and Chemical Laboratory Diagnostics (ZIMCL), Tirol Kliniken GmbH, Anichstrasse 35, 6020 Innsbruck, Austria; alexander.egger@tirol-kliniken.at; 4Central Institute for Blood Transfusion and Immunological Department, Tirol Kliniken GmbH, Anichstrasse 35, 6020 Innsbruck, Austria; harald.schennach@tirol-kliniken.at

**Keywords:** COVID-19, interferon-gamma, neopterin, biomarkers, tryptophan, phenylalanine, long COVID

## Abstract

After COVID-19, patients have reported various complaints such as fatigue, neurological symptoms, and insomnia. Immune-mediated changes in amino acid metabolism might contribute to the development of these symptoms. Patients who had had acute, PCR-confirmed COVID-19 infection about 60 days earlier were recruited within the scope of the prospective CovILD study. We determined the inflammatory parameters and alterations in tryptophan and phenylalanine metabolism in 142 patients cross-sectionally. Symptom persistence (pain, gastrointestinal symptoms, anosmia, sleep disturbance, and neurological symptoms) and patients’ physical levels of functioning were recorded. Symptoms improved in many patients after acute COVID-19 (*n* = 73, 51.4%). Still, a high percentage of patients had complaints, and women were affected more often. In many patients, ongoing immune activation (as indicated by high neopterin and CRP concentrations) and enhanced tryptophan catabolism were found. A higher phenylalanine to tyrosine ratio (Phe/Tyr) was found in women with a lower level of functioning. Patients who reported improvements in pain had lower Phe/Tyr ratios, while patients with improved gastrointestinal symptoms presented with higher tryptophan and kynurenine values. Our results suggest that women have persistent symptoms after COVID-19 more often than men. In addition, the physical level of functioning and the improvements in certain symptoms appear to be associated with immune-mediated changes in amino acid metabolism.

## 1. Introduction

The clinical manifestation of acute COVID-19 can fluctuate, depending on the variant of the virus that caused the infection, as well as comorbidities, gender, genetic susceptibility, and epigenetic modifications [1]. Similarly, patients suffering from “long haul COVID” also present with a wide range of symptoms, with intense fatigue being the most disabling one [2]. The high incidence of post COVID-19 fatigue raises concern that the infection might trigger post-viral fatigue syndrome [3].

An ongoing, but inefficient or dysfunctional, immune response has been related to chronic fatigue after infections. Immune-mediated changes in amino acid metabolism—especially tryptophan metabolism—were proposed to contribute significantly to the development of fatigue, depression, and sleep disturbances [4,5,6,7,8].

Type 1 helper cells produce the pro-inflammatory cytokine interferon-gamma (IFN-γ), which activates the guanosine triphosphate (GTP)-cyclohydrase I pathway to form either tetrahydrobiopterin (BH4) or neopterin [9]. Neopterin is produced mainly by activated macrophages and dendritic cells and is a reliable marker to assess the extent of the Th1-type immune response and the formation of reactive oxygen species (ROS) induced by IFN-γ [10,11,12,13]. BH4 is a necessary co-factor for the conversion of phenylalanine to tyrosine, and for catecholamine and serotonin synthesis. Enhanced ROS production can destroy the oxidation-labile BH4 and could therefore subsequently reduce the synthesis of dopamine and catecholamines [9].

On the other hand, elevated catecholamine concentrations might also be a problem in acute COVID-19 or long COVID; catecholamine levels vary strongly during acute infections and are also dependent on the stage of infection, age, comorbidities, and the immune status of patients, and are essential in the physiological regulation of body systems (e.g., cardiovascular, metabolic, respiratory, immune, and hematological). Additionally, catecholamine formation can also be increased by a high stress level, which can either be due to a high workload, or, more importantly, due to feelings of anxiety and helplessness [14]. During the pandemic, anxiety and depression rates rose significantly [15]. In fact, anxiety can strongly impair the immune system [16].

As catecholamines are able to diminish the T-helper cell type 1 cell-mediated cytokine response mediated by, e.g., interleukins (IL) 1, 2, 12 or IFN-γ and tumor necrosis factor alpha (TNF-α), viral replication may be facilitated in patients with COVID-19 [17]. The chronic elevation of catecholamines in patients with COVID-19, on the other hand, may increase the risk of a debilitated immune response during active infection and disturb adrenergic neurotransmitter pathways in patients with acute illness [9].

Furthermore, overwhelming catecholamine production for a longer period might finally result in a decreased ability of the adrenals to form these hormones—probably also due to the decreased availability of precursor amino acids, vitamins, and methyl groups. In line with this hypothesis, decreased concentrations of tyrosine have been demonstrated in patients 60 days after COVID-19 [13]. Additionally, enhanced immune-mediated tryptophan catabolism was found in reconvalescent patients, which may indicate that serotonin (5-hydroxytryptamin, 5-HT), kynurenine, and melatonin formation could also be affected [13]. Lowered serotonin levels go along with changes in behavior, mood, and fatigue [5,18], but also influence cognitive performance, as tryptophan deficiency is critical for brain regions associated with cognitive development like the hippocampus and the cerebellum [19]. A decrease in serotonin synthesis can possibly occur during inflammation processes, as the serotonin pathway competes with the kynurenine pathway for the substrate tryptophan [20]. Studies also show tryptophan involvement in depression through decreased levels in kynurenic acid and higher kynurenine levels, as demonstrated in patients with depressive tendencies [21].

Figure 1 shows how the pro-inflammatory cytokines, most importantly IFN-γ, induce indoleamine 2,3-dioxygenase (IDO-1) during an activated immune response. IDO-1 activity can be estimated by the calculation of the serum kynurenine to tryptophan ratio (Kyn/Trp) if other immune activation markers like neopterin are elevated in parallel [21]. Kyn/Trp can change depending on the metabolic activity. In patients with acute COVID-19, high Kyn/Trp and elevated neopterin levels were shown to be strongly related to disease severity [22,23]. A recent study also demonstrated that patients with long COVID had severe abnormalities in body temperature, increased oxidative stress, lowered antioxidant defenses, and higher total depression, anxiety, and fibromyalgia-fatigue scores [24]. Neuro-immune and neuro-oxidative pathways induced by inflammation during acute infection can also predict symptoms like chronic fatigue, depression, and anxiety [25]. In this study, we investigated symptom persistence and inflammation-mediated changes in aromatic amino acid metabolism in patients recovering from COVID-19.

## 2. Materials and Methods

### 2.1. Participants and Study Design

Patients who had had acute, PCR-confirmed COVID-19 about 60 days earlier were recruited within the scope of the prospective CovILD study that was carried out at the outpatient ward of the Internal Medicine II department of Innsbruck University Hospital from April to July 2020. A total of 142 participants aged 19 to 86 recovering from COVID-19 were included. As blood specimens from acute illness were only available from less than a third of patients (Gietl et al. [13]), we performed cross-sectional analyses, investigating the relationship between persistent symptoms and biochemical alterations of amino acid metabolism. Each patient signed a written consent form that his/her blood and/or urine samples may be used for scientific purposes. The institutional review board commission of the Medical University Innsbruck approved the study protocol and the storage of patient samples in a biobank (1157/2017 and 1103/2020) and the study was registered at ClinicalTrials.gov (registration number NCT04416100). The relevant data were anonymized through the allocation of a sample number from P001-P142. Patients were followed-up approximately 60 days after the onset of symptoms, as described earlier [26]. During the follow-up (FU) appointment, patients answered a questionnaire, and both blood sampling and clinical/pneumological evaluation were performed. 

### 2.2. Data Collection

Demographic information was collected from participants and further information was retrieved from patient records, which included inpatient admission, age and gender, BMI, complete and differential blood count, serum glucose levels, infectious parameters, electrolytes, as well as cardiac markers and kidney function parameters. The need for intubation/oxygenation and anti-infectious therapy of the participants was noted. To assess the patients’ physical level of functioning and their ability to care for themselves, the Eastern Cooperative Oncology Group (ECOG) performance status score was used. Clinical presentation (ECOG 0–4, pain, gastrointestinal symptoms, anosmia, sleep quality and other neurological symptoms) during acute COVID-19 and FU were recorded using questionnaires.

### 2.3. Laboratory Analysis

The blood sampling was performed at the time of the FU. Routine laboratory values were analyzed by the ISO-15189-accredited Central Institute for Medical and Chemical Laboratory Diagnostics (ZIMCL) in Innsbruck, Austria, as follows: CRP, IL-6, iron, transferrin, ferritin, sTfR, folate, Vit B12, high sensitivity TropT, CK, NT-proBNP, and enzymatic creatinine were analyzed using a Cobas 8000 platform (Roche Diagnostics, Rotkreuz, Switzerland). Bioactive Hepcidin was determined using ELISA (DRG Instruments GmbH, Marburg, Germany) on a BEP2000, and 25-OH Vit D was measured via HPLC using the kit from Chromsystems Instruments and Chemicals GmbH (Graefelfing, Germany). All hematologic parameters were determined using an XN-2000 analyzer from Sysmex (Kobe, Japan).

Neopterin and amino acid measurements were performed at the Institute of Medical Biochemistry, Biocenter, of the Medical University of Innsbruck. Neopterin concentrations were measured using an enzyme-linked immunosorbent assay (BRAHMS GmbH, Hennigsdorf, Germany) following the manufacturer’s protocol (sensitivity: 2 nmol/L). Concentrations of kynurenine, tryptophan, phenylalanine, and tyrosine were analyzed in the patients’ plasma via high-performance liquid chromatography, as described earlier [13]. Nitrite content was determined using a modified Griess reaction, as described earlier [27]. In brief, after chromatographic separation on a LiChroCART RP-18 endcapped column (55–4.3 μm, Merck, Darmstadt, Germany), the analytes tryptophan, phenylalanine, and tyrosine were measured via their native fluorescence (tryptophan: excitation wavelength of 286 nm, emission wavelength of 366 nm; phenylalanine and tyrosine excitation wavelength of 210 nm, emission wavelength of 302 nm). Kynurenine and the internal standard, L-nitrotyrosine, were detected via UV absorbance at a wavelength of 360 nm. The ratios of kynurenine to tryptophan (Kyn/Trp) and phenylalanine to tyrosine (Phe/Tyr) were calculated as surrogates of indoleamine 2,3-dioxygenase and phenylalanine hydroxylase activity, respectively [27,28,29].

### 2.4. Questionnaire

A detailed questionnaire was handed out to the patients before the clinical investigation; data for acute COVID-19 were obtained retrospectively. Quality of sleep was assessed through the patients’ own statements, the prescription of sleep medication, or both. To analyze how the disease was impacting the patients’ general functioning ability, the ECOG performance scale was used for standardization and reference purposes. If a discrepancy was observed regarding the patients’ statements and their clinical examination, then the data of the structured medical interview were chosen.

### 2.5. Additional Data and Definitions

Data on depression and antidepressant intake were acquired from the patients’ medical files to investigate whether the symptoms were diagnosed prior to COVID-19 or during acute illness, or at FU. Furthermore, data on a broad spectrum of neurological symptoms like dizziness, sensory disturbance, headache, and cognitive impairment were gathered.

### 2.6. Statistical Evaluation

A cross-sectional analysis of the investigated lab parameters was performed. Furthermore, the percentage of patients with persistent symptoms was calculated. All statistical analyses were carried out using the statistical analysis software package IBM SPSS Statistics (version 27.0.1.0. by IBM Corporation and its licensors 1989, 2020, Armonk, NY, USA). *p*-values < 0.05 were considered statistically significant. Correlations were evaluated using the Spearman rank test. The frequency and percentage of categorical variables were evaluated. To describe non-normally distributed continuous variables, medians and interquartile ranges were used. As most of the variables were non-normally distributed according to Kolmogorov–Smirnov, non-parametric tests were applied. Mann–Whitney-U was used to compare two independent groups, whereas Kruskal–Wallis compared variables of more than two independent groups. A chi-square test was applied to analyze group differences in qualitative variables.

Amino acid and neopterin concentrations were compared to the already available data of healthy blood donors, which had been published earlier [27].

## 3. Results

### 3.1. Baseline Characteristics of the Study Population

In total, 142 patients (63 women, 44.4%, and 79 men, 55.6%) were included in the study. All of them suffered from acute COVID-19 approximately 60 days earlier and revisited our clinic for a follow up (FU) investigation at the out-patient clinic, Department of Internal Medicine II of Innsbruck University Hospital in Innsbruck, Austria. Overall, 94 (66.2%) patients had been treated as in-patients and the remaining 48 (33.8%) as out-patients during acute COVID-19. The mean age was 57.3 ± 14.1 years, ranging from 19.0 to 87.0 years, and the mean BMI was 26.4 ± 4.8 kg/m^2^.

Most individuals had pre-existing comorbidities, the most frequent being cardiovascular and metabolic diseases. Only 31 patients had no chronic systemic diseases (pulmonary disease, cardiovascular disease, endocrinologic or gastrointestinal disease, malignant or chronic kidney disease, immunodeficiency) according to their medical records. Fifty-eight patients had at least one or more cardiovascular co-morbidities (i.e., obesity, hypertension, diabetes, coronary artery disease/stroke, hyperlipidemia), and twenty-seven had pre-existing pulmonary diseases like asthma, COPD, or interstitial lung disease. During acute COVID-19, 64 patients had been treated with oxygen, 24 with invasive/non-invasive ventilation, and 80 had received anti-infectious treatment.

### 3.2. Laboratory Parameters

In Table 1, the routine laboratory parameters and additional measurements of plasma tryptophan, kynurenine, phenylalanine, tyrosine, neopterin, and nitrite are shown for the whole population as well as for men and women separately. Most of the investigated parameters did not differ between men and women (see also means ± SEM in Table 1). Only the parameters of iron metabolism, vitamin B12, vitamin D, troponin T, creatine kinase, creatinine, folate, and glucose differed significantly between men and women (bold in Table 1).

### 3.3. Clinical Symptoms during Acute COVID-19 and at FU

The frequency of patients’ symptoms during acute COVID-19 and FU is shown in Table 2. Interestingly, the symptom load at FU was much higher in female patients. Male patients reported less symptoms (except for neurological symptoms) during acute COVID-19 and at FU, while female patients had significantly more persistent symptoms, specifically gastrointestinal malaise (GI; *p* = 0.038) and anosmia at FU (*p* = 0.002). An improvement in the quality of sleep at FU was only stated by male patients, while the symptom remained unchanged in women (*p* = 0.031). Female patients also tended to have persistent pain more frequently than men (*p* = 0.093).

### 3.4. Inflammation and IFN-γ-Mediated Biochemical Pathways in Patients at FU

IFN-γ-mediated biochemical pathways were still activated in many patients after 60 days of symptom onset.

A high percentage of patients presented with elevated levels of CRP (76%) and neopterin (>9.1 nM, *n* = 80, 56%), while increased interleukin-6 (IL-6) was only found in 15 patients. Compared to the values of healthy blood donors [27], tryptophan and tyrosine levels were lower than the 95th percentile in many patients (*n* = 70 for tryptophan, *n* = 65 for tyrosine), while phenylalanine was higher than the 95th percentile in 26 patients. Increased Phe/Tyr and Kyn/Trp ratios (Phe/Tyr: *n* = 111; Kyn/Trp: *n* = 112) indicated a high turnover of these amino acids in the regeneration phase of COVID-19. Thus, high percentages of patient levels of the investigated parameters were out of the reference range, indicating ongoing immune activation processes.

The values of tryptophan, kynurenine, neopterin, phenylalanine, and tyrosine did not differ significantly between men and women in our cohort. However, as gender differences were described earlier in healthy individuals [27], and women were affected more often by symptom persistence, we also performed gender-stratified analyses in which we compared levels of these parameters with levels measured earlier in healthy blood donors [27]. All parameters (except for nitrite in women) differed significantly compared to the respective control group (all *p* < 0.001 for men and for women, see also Figure 2).

A higher neopterin coincided with enhanced IDO-1 activity (as reflected by Kyn/Trp [rs = 0.708, *p* < 0.001, see Figure 3]), indicating enhanced tryptophan catabolism following immune activation. In men, higher tyrosine and phenylalanine levels were found in patients with higher nitrite levels (tyrosine: rs = 0.371, *p* < 0.001; phenylalanine: rs = 0.308, *p* = 0.006).

Patients with low tryptophan concentrations also presented with low phenylalanine (rs = 0.412, *p* < 0.001) and tyrosine (rs = 0.638, *p* < 0.001) levels (see Figure 3), which might be due to an increased turn-over of amino acids during acute infection and reconvalescence. In line with enhanced consumption, low levels of vitamin D (data presented elsewhere [26]), B-vitamins (see Table 1), and iron (see also [30]) were also encountered frequently.

Patients without any of the afterwards mentioned comorbidities (obesity, diabetes, hypertension or other cardiovascular, pulmonary, malignant, endocrine, or gastrointestinal comorbidities, or immunodeficiency; *n* = 31) presented with lower neopterin concentrations (median 8.2 nM vs. 9.7 nM) compared to patients who had at least one of these comorbidities. All other investigated lab parameters did not differ. The number of comorbidities did not correlate with the investigated lab parameters, neither in the whole cohort nor in women or men. Lower neopterin concentrations were found in men without co-morbidities (*n* = 17). In women with comorbidities (*n* = 14), neopterin levels were similar to patients without co-morbidities. Patients with cardiovascular comorbidities (i.e., at least one of the following: obesity, diabetes, coronary artery disease/stroke, hypertension, or hyperlipidemia; n = 58) had higher neopterin values (median 10.3 vs. 9.5 nM) and lower nitrite levels (median 18.5 vs. 26.7 µM), but higher Phe/Tyr ratios (1.19 vs. 1.12).

### 3.5. Correlation of Clinical Symptoms and IFN-γ-Mediated Biochemical Pathways in Patients after COVID-19

Overall, 69 patients were asymptomatic at FU (vs. *n* = 35: 1 symptom; *n* = 28: 2 symptoms; *n* = 10: ≥ 3 symptoms). Creatinine kinase (CK) was significantly lower in asymptomatic vs. symptomatic patients (*p* = 0.006); all other parameters did not differ.

The physical performance of female patients (as depicted by the ECOG score) was associated with Phe/Tyr. Women with a lower physical performance (e.g., higher ECOG scores) presented with an increased Phe/Tyr (*p* = 0.035, ⴄ = 0.3661, see also Figure 4). No such association was found in men.

For each symptom, a variable was created to analyze the difference and improvement in each complaint stated during COVID-19 reconvalescence. An increment of ≥2 in the ECOG scoring was defined as a significant improvement. No improvement was reported by 50 (35.2%, *n* = 22 female, n = 28 male) patients, whereas 92 (64.8%, *n* = 39 female, *n* = 53 male) patients reported ECOG improvement. Female patients whose physical abilities improved after COVID-19 (as indicated by an ECOG change ≥ 2) tended to have lower Phe/Tyr (*p* = 0.087), whereas no differences were observed in men.

Patients who reported less pain (*n* = 45) presented with a decreased Phe/Tyr ratio (*p* = 0.038). Tyrosine concentrations were higher in females with improving pain (*n* = 25, *p* = 0.049), and Phe/Tyr was lower (*p* = 0.028). As expected, pain during acute COVID-19 was more strongly affecting older patients (ⴄ = 0.635).

Patients with improved gastrointestinal symptoms during reconvalescence (*n* = 48) presented with higher kynurenine (*p* = 0.044) and tryptophan (*p* = 0.050) values. In patients whose anosmia, sleep disturbance, and neurologic and/or psychiatric symptoms improved, IFN-γ-mediated parameters were similar to the values of patients with persistent symptoms.

## 4. Discussion

The results of our cross-sectional analysis show that in most reconvalescent COVID-19 patients, IFN-γ-mediated biochemical pathways were still strongly activated after 60 days. Elevated levels of plasma neopterin and CRP were found in more than the half of patients indicating the hyperactivation of the immune system, especially Th1-type immune activation. Kynurenine and the ratios of kynurenine to tryptophan and phenylalanine to tyrosine were elevated in a high percentage of patients, indicating a high turnover of these amino acids in the regeneration phase of COVID-19. Serum tryptophan was low in many patients, and the same was true for tyrosine.

We could demonstrate symptom improvement in many patients during reconvalescence, but a high percentage of patients still had complaints after 60 days. Women suffered from symptom persistence more often, which fits well with other studies showing an association of female gender with long COVID/post-COVID syndrome [31,32]. One can only speculate about the possible reasons for this, and a higher susceptibility to stress-related sleep and anxiety disorders through emotional stress has been proposed [33]. In fact, the prevalence of anxiety, depression, and emotional stress in the general population increased strongly during the lockdown, and women were found to be more burdened [15]. Additionally, psychosocial stress was present amongst other variables like physical performance loss. High numbers of acute and sub-acute COVID-19 as well as neurocognitive symptoms were shown to be some of the strongest correlates of deteriorating mental health and quality of life in patients after COVID [34] in an online survey, in which more than 60% were female participants.

Psychosocial stress and anxiety might also predispose patients to have a worse outcome or more symptoms during acute and reconvalescent COVID-19. In line with this hypothesis, pre-existing depression and anxiety were associated with an increased symptom burden during and after COVID-19; however, the deterioration of mental health in non-hospitalized COVID-19 convalescents was quite notable. Anxiety increased from 6% to 12.4% in Austrian individuals, and from 4.6% to 19.3% in Italian individuals, while depression was found in 17.3% of Austrian and 23.2% of Italian convalescents in one binational online survey [34]. Unfortunately, gender differences were not reported in this cohort.

In fact, anxiety, depression, and psychosocial stress might impact patients’ immune response against the virus significantly. Stressful life events have a well-established decreasing effect on immunity (see also review by G. Schüssler, C. Schubert) [16]. As the lockdown and pandemic situation were very stressful for most people, emotional/mental stress probably had a significant impact on the patients’ ability to respond properly to the infection. Additionally, sex-dependent differences in the immune response to SARS-CoV-2 infection have probably been implicated in the COVID-19 gender imbalance [35].

Men presented with higher cardiac markers and iron parameters, which confirms earlier data [30,36,37]. Hyperferritinemia and anemia have been demonstrated in this cohort after 60 days [30]. Disturbances in iron metabolism coincided with low B vitamin levels during reconvalescence, possibly also contributing to anemia, which was found in six patients. Interestingly, values of folate, vitamin B12, and vitamin D were significantly higher in female patients, suggesting that men either had less consumption of these important vitamins or that women—which is more probable—had a higher intake, either through their diet or supplements.

Nearly all patients had decreased vitamin D levels, which also has been shown earlier [26], and might be related to increased consumption during infection, seasonal fluctuations, or may have been amplified by lockdown regulations. Deficiencies or low levels of vitamins and amino acids during reconvalescence in general might be caused by ongoing and overwhelming inflammation/immune processes. Associations between high neopterin levels and Kyn/Trp indicate an enhanced turnover of essential nutrients, and that symptom persistence/improvement might be related to biochemical changes. Interestingly, a lower level of functionality (i.e., a higher ECOG score) was seen in female patients with higher Phe/Tyr. Women also tended to have lower Phe/Tyr when their ECOG scores improved. Patients (especially women) who reported improvements in pain during reconvalescence also presented with lower Phe/Tyr compared to patients without amelioration. Similarly, tryptophan and kynurenine levels were higher in patients in whom gastrointestinal symptoms were improved.

Gastrointestinal symptoms often coincided with other symptoms like anosmia and sleep disturbances (which persisted in more than 25% of patients). All women who had sleep problems during acute COVID-19 still could not sleep well at FU, while only 14 of the 26 initial male patients still reported sleep disturbances. Interestingly, we did not find an association between sleep disturbance and tryptophan levels at FU, while low tryptophan concentrations and higher inflammatory markers were associated with sleep disturbance during acute COVID-19 [13]. During acute COVID-19, earlier metabolic profiling showed lower tryptophan and higher kynurenine, neopterin, and phenylalanine levels compared to healthy individuals, as well as correlations between the key metabolite changes and concentrations of proinflammatory cytokines and chemokines [38]. Thus, immunometabolic changes appear to occur not only during acute infection [38], but also thereafter. In fact, persistent symptoms/physical limitations are related to certain metabolic changes, but also seem to be associated with different symptoms. Very often, patients with persistent symptoms also had several symptoms simultaneously. Patients with decreased sleep quality also suffered from anosmia, pain, and gastrointestinal malaise at FU quite frequently. In total, 24 patients had persistent neurological symptoms at FU. Patients with a higher ECOG score were more severely affected.

The limitations of this study are that symptoms during acute COVID-19 were gathered retrospectively via non-systematic data gathering and were not assessed before the onset of illness. Therefore, the self-constructed questionnaires applied in this study might lack the usual quality criteria (objectivity, reliability, validity). Further, the study includes only the data of one follow-up, approximately 60 days after acute illness, where the blood samples were also taken. As blood specimens from acute illness were only available from less than a third of the patients (Gietl et al. [13]), we only performed a cross-sectional analysis using the data of 142 patients.

Therefore, itis also quite difficult to extrapolate to which extent the biomarker alterations were not already pre-existing (e.g., due to other concomitant diseases) or related to psychological triggering factors like stress and anxiety. We tried to account for this fact by further calculations, in which we investigated whether patients who had co-morbidities, according to their records, differed regarding interferon-gamma-mediated biochemical pathways in comparison to patients who had no known co-morbidities. In fact, neopterin concentrations were higher in patients who had comorbidities or pre-existing cardiovascular co-morbidities, and also Phe/Tyr was higher if patients had at least one cardiovascular comorbidity. Thus, not all “post-COVID” symptoms that were correlated with inflammation cross-sectionally might be attributable to COVID-19 and/or to the underlying disease.

Longitudinal studies investigating these questions in a bigger cohort with a more homogenous population and defined pre-existing comorbidities (or chronic immune-related issues) should therefore be conducted in order to obtain a more exact picture of the underlying pathomechanisms.

## 5. Conclusions

To summarize, our data suggest that the ongoing activation of IFN-γ-mediated pathways might influence the further course of reconvalescence; continuous immune activation might go along with an enhanced demand for nutrients like amino acids and vitamins. Furthermore, we could demonstrate gender-specific differences regarding symptom load and development. Therefore, it might be interesting to further investigate these results in larger cohorts, and longitudinally track biomarker alterations during convalescence.

However, it is also very important to look at these biomarkers in the right context, as one patient is not like another and every patient has different genetic, environmental, and psychosocial strengths. Additionally, impairments, individual resources, and resilience should also be considered. Integrative health approaches, taking into account the bio-psychosocial model and new mind–body interventions to balance psychoneuroimmunological circuits [39], might in fact be very helpful to improve the outcomes of patients.

## Figures and Tables

**Figure 1 jpm-13-01055-f001:**
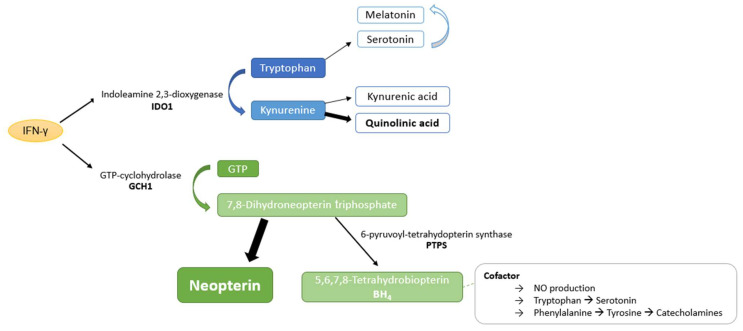
Inflammation-associated biochemical pathways. Inflammatory signaling, most importantly interferon gamma (IFN-γ), stimulates neopterin synthesis via GTP cyclohydrolase 1 (GTP-CH-I) and tryptophan (Trp) catabolism along the kynurenine (Kyn) axis. Neopterin formation occurs mainly in human macrophages (MΦ) and dendritic cells (DC) at the expense of tetrahydrobiopterin (BH4). BH4 is a cofactor of monoxygenases, e.g., phenylalanine 4-monooxygenase (PAH), tyrosine 3-monooxygenase (TH), tryptophan 5-monooxygenases (TPH), and nitric oxide synthases (NOS). BH4 can be synthetized by other cell types, but it is oxidation labile and may diminish in situations of oxidative stress. Abbreviations: KynA = kynurenic acid, indoleamine 2,3-dioxygenase 1 (IDO-1), NAD = nicotinamide adenine dinucleotide, Phe = phenylalanine, QuinA = quinolinic acid, Tyr = tyrosine. Metabolites analyzed in the study are in bold and highlighted grey. Dashed arrows indicate biosynthetic processes in which more than one step/enzyme is involved. Figure licensed by Talia Piater.

**Figure 2 jpm-13-01055-f002:**
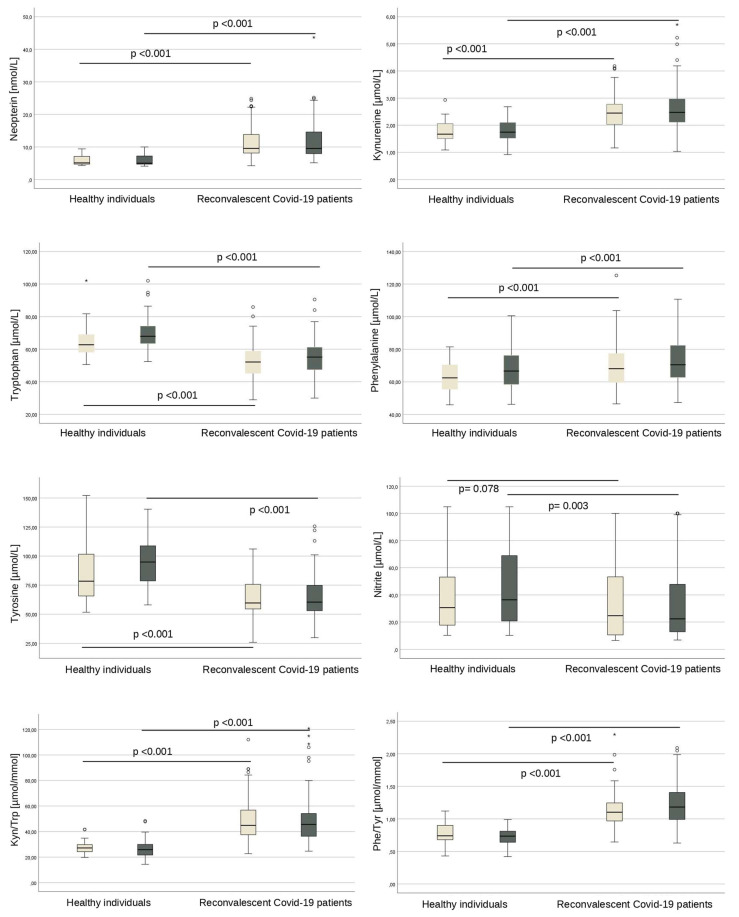
Gender comparison of IFN-γ-mediated parameters. All parameters except nitrite (*p* = 0.003 for men and *p* = 0.078 for women, n.s.) are significantly different between healthy blood donors and reconvalescent COVID-19 patients, both in women (light boxplots) and in men (dark grey boxplots), with *p* < 0.001 in each case. * and ° outlier.

**Figure 3 jpm-13-01055-f003:**
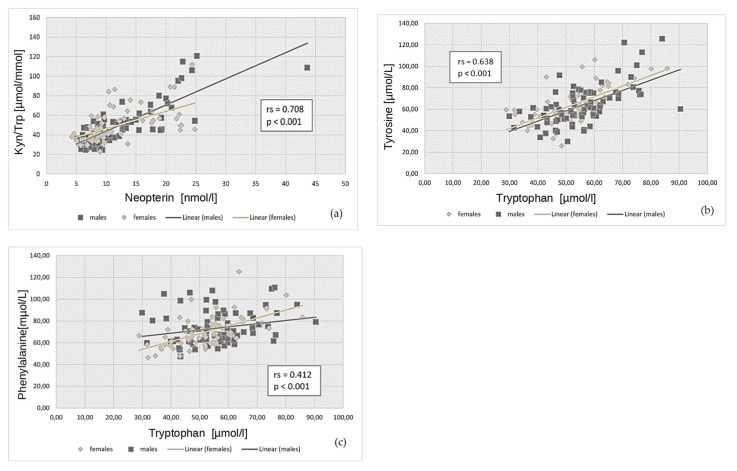
Scatter plots demonstrating the relationship between selected amino acids and/or ratios stratified by gender. Kyn/Trp = kynurenine to tryptophan ratio. (**a**) Higher neopterin values correlated with enhanced IDO-activity (as reflected by Kyn/Trp). (**b**) Low tryptophan values correlated significantly with low tyrosine levels. (**c**) Patients with low tryptophan concentrations also presented with low phenylalanine.

**Figure 4 jpm-13-01055-f004:**
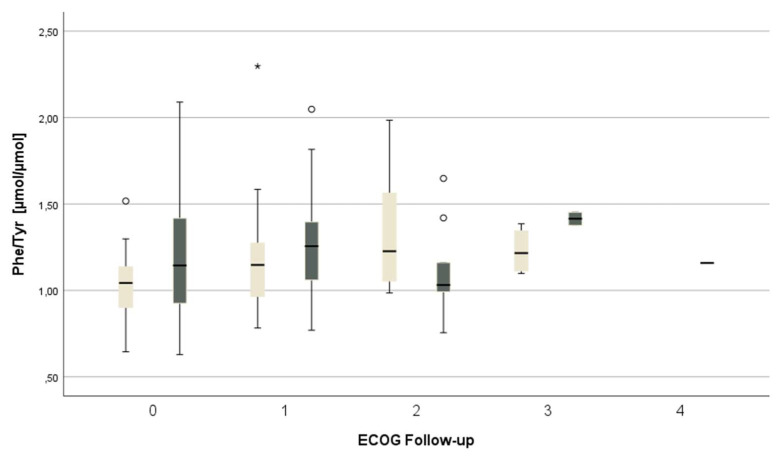
Gender comparison of ECOG score and Phe/Tyr ratio. Women = light, men = dark. Lower physical performance of females correlated positively with higher Phe/Tyr values. No significant correlation was found in men. Phe/Tyr = phenylalanine to tyrosine ratio. * and ° outlier.

**Table 1 jpm-13-01055-t001:** Routine laboratory parameters, inflammatory parameters, and markers of amino acid metabolism at follow-up (FU) with mean +/− SEM and reference value (central 95%). The corresponding *p*-value was computed while controlling for gender; n.s. = non-significant.

		Gender [f/m]			
		Female (*n* = 63)	Male (*n* = 79)		
Laboratory Values at Follow-Up	Mean (SEM)	Mean (SEM)		Reference Value	*p*-Value
Neopterin ^1^ (nM/L)	11.9 (0.5)	11.7 (0.7)	12.2 (0.7)	5.9 ± 1.6	n.s.
Nitrite ^1^ (µM/L)	36.1 (2.6)	36.7 (4.1)	35.5 (3.3)	44.9 ± 32.0	n.s.
Kynurenine ^1^ (µM/L)	2.56 (0.07)	2.47 (0.08)	2.64 (0.10)	1.78 ± 0.42	n.s.
Tryptophan ^1^ (µM/L)	54.08 (0.99)	52.29 (1.46)	55.50 (1.33)	67.4 ± 10.2	n.s.
Phenylalanine ^1^ (µM/L)	71.98 (1.22)	70.05 (1.8)	73.53 (1.66)	65.2 ± 11.1	n.s.
Tyrosine ^1^ (µM/L)	63.89 (1.48)	64.14 (2.13)	63.69 (2.07)	90.6 ± 22.9	n.s.
Kyn/Trp ^1^ (µM/mM)	49.46 (1.63)	49.35 (2.19)	49.56 (2.36)	26.7 ± 6.2	n.s.
Phe/Tyr ^1^ (µM/µM)	1.18 (0.03)	1.14 (0.03)	1.22 (0.04)	0.75 ± 0.14	n.s.
CRP (mg/dL)	0.38 (0.09)	0.31 (0.06)	0.45 (0.15)	<0.5	n.s.
IL-6 (ng/L)	3.8 (0.5)	3.9 (0.9)	3.7 (0.7)	<7.0	n.s.
WBC (G/L)	6.45 (0.19)	6.32 (0.23)	6.55 (0.29)	4.0–10.0	n.s.
Neutrophils %	3.86 (0.16)	3.78 (0.21)	3.92 (0.23)	46.0–66.0	n.s.
Hemoglobin (g/L)	137.51 (1.26)	132.32 (1.37)	141.66 (1.85)	120.0–180.0 ^2^	**<0.001**
Thrombocytes (G/L)	260.98 (6.30)	268.24 (8.19)	255.19 (9.25)	150.0–380.0	n.s.
Iron (µM/L)	15.6 (0.5)	15.2 (0.6)	16 (0.7)	5.8–34.5	n.s.
Transferrin (mg/dL)	249 (3.0)	253 (6.0)	246 (4.0)	200.0–360.0	n.s.
Ferritin (µmol/L)	262 (20)	167 (23.0)	339 (29.0)	15.0–400.0 ^2^	**<0.001**
Transferrin saturation (%)	25.0 (1.0)	25 (1.0)	26 (1.0)	16.0–45.0	n.s.
sTfR (mg/L)	3.4 (0.1)	3.2 (0.1)	3.5 (0.1)	1.80–4.7 ^2^	**0.046**
Hepcidin-25 (µg/L)	20.6 (1.4)	17 (1.7)	23.6 (2.0)	1.5–41.5	**0.015**
Folate (μg/L)	7.41 (0.36)	7.02 (0.46)	7.72 (0.54)	3.9–26.8	**0.017**
Vit B12 (pg/mL)	292.18 (20.42)	303 (14.0)	292 (20.0)	145.0–569.0	**0.050**
25-OH Vit D (nM/L)	55 (2.0)	60 (3.0)	50 (2.0)	75.0–150.0	**0.005**
Troponin T (ng/L)	10.1 (0.7)	7.2 (0.6)	12.4 (1.1)	<14.0	**<0.001**
Creatine kinase (CK, U/L)	92 (5.0)	81 (8.0)	102 (7.0)	26.0–190.0	**0.006**
NT-proBNP (ng/L)	243 (45)	161 (26.0)	309 (77.0)	<486.0 ^2^	n.s.
Creatinine (mg/dL)	0.84 (0.02)	0.74 (0.02)	0.92 (0.03)	0.51–1.17	**<0.001**

^1^ Reference ranges for neopterin, nitrite, kynurenine, tryptophan, phenylalanine, tyrosine, kynurenine/tryptophan, and phenylalanine/tyrosine were obtained from Geisler et al. [27]. ^2^ Gender-dependent reference values for hemoglobin: 120.0–157.0 g/L for women; 130.0–177.0 g/L for men; ferritin: 15.0–150.0 µg/L for women 18–50 years, 30.0–400.0 µg/L for women 51–120 years; 30.0–400.0 µg/L men 18–120 years; sTfR: 1.78–4.59 mg/L for women; 1.80–4.70 mg/L for men; CK: 26.0–170.0 U/L for women; 39.0–190.0 U/L for men; NT-proBNP: ≤202.0 ng/L for women 18–45 years, ≤226.0 ng/L until 55 years, ≤284.0 ng/L until 65 years, ≤470.0 ng/L until 120 years; ≤90.80 ng/L for men 18–45 years, ≤121.0 ng/L until 55 years, ≤262.0 ng/L until 65 years, ≤486.0 ng/L until 120 years; creatinine: 0.51–0.95 mg/dL for women; 0.67–1.17 mg/dL for men.

**Table 2 jpm-13-01055-t002:** Frequency of symptom occurrence in patients during acute COVID-19 and at follow-up. Significant gender differences in symptom occurrence are shown by bold letters. Abbreviations: *n* = number of patients; GI COVID = Gastrointestinal symptoms during COVID-19; n.s. = non-significant.

	Total (*n* = 142)	Female (*n* = 63)	Male (*n* = 79)	*p*-Value (f/m)
	*n* (Row Valid %)			
Pain COVID	**71 (50.0)**	37 (58.7)	34 (54.4)	**0.028**
Follow-up	26 (18.3)	15 (23.8)	11 (13.9)	n.s.
GI COVID	58 (40.8)	31 (49.2)	27 (34.2)	**0.036**
Follow-up	11 (7.7)	8 (12.7)	3 (3.8)	**0.038**
Anosmia COVID	59 (41.5)	37 (58.7)	22 (27.8)	**<0.001**
Follow-up	20 (14.1)	15 (23.8)	5 (6.3)	**0.002**
Sleep COVID	48 (33.8)	22 (34.9)	26 (32.9)	n.s.
Follow-up	36 (25.3)	22 (34.9)	14 (17.7)	**0.031**
Neuro COVID	61 (42.9)	27 (42.9)	34 (43.0)	n.s.
Follow-up	23 (16.2)	9 (14.3)	14 (17.7)	n.s.

## Data Availability

Results presented within this study are available within the manuscript. Blinded raw data are available on demand.

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
