# Peer review of "Persistent Symptoms and IFN-γ-Mediated Pathways after COVID-19"

_jpm, 2023, doi:10.3390/jpm13071055_

Round 1

Reviewer 1 Report

Persistent symptoms and IFN-γ mediated pathways after COVID-19

This study compared inflammatory parameters and alterations of tryptophan and phenylalanine metabolism in 142 patients during an acute COVID-19 manifestation and approx. 60 days later while 69 of them still suffered from long COVID symptoms.

Abstract: Design, methods and findings should be worked upon, not clearly written. For example, you do not mention here that this study is a cross-sectional analysis of a prospective study.

Introduction, first paragraph: The various intensities and clinical manifestations of COVID as well as long COVID can be also related to psychological, i.e. stress factors (anxiety, post-trauma). Please include. Stress is triggering inflammation (neopterin, kyn/trp) is triggering fatigue (sickness behavior).

Line 60: Again, catecholamine release is highly dependent from stress and anxiety. Leaving out this connection produces a biased “reductionistic” picture towards the virus as a sole factor connected with acute COVID and particularly long COVID.

Line 148: You mention a detailed questionnaire. However, it remains unclear what the questionnaire comprises. Please provide reference or – if self-constructed – report on the validity and reliability of the research instrument. Moreover, you speak of a possible discrepancy between patients' statements and their clinical examination. What does that mean? Please specify. Please also specify the structured medical interview, which was used when the discrepancy emerges.

Line 158: How were data on a broad spectrum of neurological symptoms like dizziness, sensory disturbance, headache, and cognitive impairment gathered? Please specify. Do you have a reference to the questionnaire or can you provide its validity and reliability data?

Line 183: You state “Most individuals had pre-existing comorbidities, the most frequent being cardiovascular and metabolic diseases.” As these diseases are usually connected with chronic immune alterations and comorbidities such as depression, fatigue etc., I am wondering how these comorbidities are controlled for in your analyses of COVID as well as long COVID.

Line 311: Why is it more likely that sex-dependent differences in immune responses to SARS-CoV-2 infection are responsible for the finding that women suffered from symptom persistance more often and not psychological factors like anxiety etc.?

Line 258: Figure 3B, missing information: rs = 0.638, p < 0.001

Line 296: Principally, there is not enough discussion of psychological stress factors as possible COVID related triggers of inflammation and sickness behavior. The authors focus on material factors such as nutrition, vitamins, amino acids and the like. This might be only a small part of the factors responsible for the differences seen.

Minor editing of English language required. Several typos.

Author Response

Dear Reviewers, 

First, we would like to thank you for taking the time to review our manuscript and for providing valuable suggestions on how to refine the manuscript for publication. Your effort and comments are very much appreciated! We revised our manuscript carefully according to the recommendations given, please find below our reply:

Abstract: Design, methods and findings should be worked upon, not clearly written. For example, you do not mention here that this study is a cross-sectional analysis of a prospective study.

We have revised the manuscript according to the reviewer’s recommendation.

Introduction, first paragraph: The various intensities and clinical manifestations of COVID as well as long COVID can be also related to psychological, i.e., stress factors (anxiety, post-trauma). Please include. Stress is triggering inflammation (neopterin, kyn/trp) is triggering fatigue (sickness behavior).

Thank you for raising this important point. We added this information in the manuscript (and revised the "Introduction" and "Discussion" to further discuss this issue. 

Line 60: Again, catecholamine release is highly dependent from stress and anxiety. Leaving out this connection produces a biased “reductionistic” picture towards the virus as a sole factor connected with acute COVID and particularly long COVID.

That’s definitely true, stress and anxiety levels are often high in patients with Long COVID - we also have analyzed data from 97 patients of this cohort, in which anxiety, fatigue and PTSD were investigated after 3 months by questionnaires: and actually, these so far unpublished data show, that  many patients had anxiety, depression or a history of trauma (HADS-A >7  23.7 %, HADS-D>7 11.3%)  and PCL-5 >32 (11.3 %). Only 2 patients had a diagnosis of depression before COVID according to the medical records, one patient suffered from an adaptive disorder with depressive and anxious components before COVID-19. As we did not have data of all patients and questionnaires were filled in about one month later, we did not include these data in this manuscript. However, we added a paragraph, in which we discuss this topic in the "Introduction".

Line 148: You mention a detailed questionnaire. However, it remains unclear what the questionnaire comprises. Please provide reference or – if self-constructed – report on the validity and reliability of the research instrument. Moreover, you speak of a possible discrepancy between patients' statements and their clinical examination. What does that mean? Please specify. Please also specify the structured medical interview, which was used when the discrepancy emerges.

The questionnaire was self-constructed and patients merely reported, whether they had symptoms or not. As patients often had different neurological symptoms during acute and reconvalescent COVID-19, the most common symptoms were queried, and patients had the possibility to also add other symptoms.

Regarding the discrepancy: If patients reported not to have neurological symptoms, but e.g., memory deficits or other neurological or dyspnea symptoms were apparent in the structured medical interview, patients were regarded to have these symptoms. However, this was only the case in single patients.

Line 158: How were data on a broad spectrum of neurological symptoms like dizziness, sensory disturbance, headache, and cognitive impairment gathered? Please specify. Do you have a reference to the questionnaire, or can you provide its validity and reliability data?

The patients were asked in the self-constructed questionnaire, whether they had the symptoms or not. Some patients had already been seen by a neurologist, but most patients not. As the self-constructed questionnaire was only used to document the presence/absence of symptoms and current medication, we cannot provide validity and reliability data.

Line 183: You state “Most individuals had pre-existing comorbidities, the most frequent being cardiovascular and metabolic diseases.” As these diseases are usually connected with chronic immune alterations and comorbidities such as depression, fatigue etc., I am wondering how these comorbidities are controlled for in your analyses of COVID as well as long COVID.

Basically, a high percentage of patients in our cohort had more than one comorbidity - which was often also the cause for hospitalization during acute COVID-19. As we could not verify every history of secondary patient diagnosis, we checked the patients’ medical records, whether they had pre-diagnosed depression or fatigue or not. Only 2 patients had depression before COVID-19 and one patient suffered from an adaptive disorder with depressive and anxious components before - at least according to the medical records of patients.

To address your question about other cardiovascular and metabolic comorbidities we also performed additional analyses, which demonstrated that patients with diabetes mellitus presented with higher phenylalanine concentrations than patients without diabetes (median 73.9 µM vs. 68.9 µM, p <0.05)- we added this new result to the manuscript. Interestingly, the other parameters did not differ significantly. Thank you for this good point, which we addressed in the "Results".

Line 311: Why is it more likely that sex-dependent differences in immune responses to SARS-CoV-2 infection are responsible for the finding that women suffered from symptom persistence more often and not psychological factors like anxiety etc.?

We have questionnaire data regarding anxiety of 97 patients of this cohort gained 3 months after COVID-19, which showed that men were affected stronger by anxiety and depression than women (HADS-A >7; men:28.1 %, women: 17.5 %; HADS-D >7 men 14 % vs. women 7.5 %). We would rather have expected that women would suffer from anxiety more frequently than men… And during the structured medical interview, many patients reported the feeling of anxiety was new to them and had begun during acute COVID.

Line 258: Figure 3B, missing information: rs = 0.638, p < 0.001

We added this information to the figure.

Line 296: Principally, there is not enough discussion of psychological stress factors as possible COVID related triggers of inflammation and sickness behavior. The authors focus on material factors such as nutrition, vitamins, amino acids, and the like. This might be only a small part of the factors responsible for the differences seen.

It is certainly true, that psychological factors play an important role before, during and after infection: Anxiety can dampen the immune response and thus facilitate infection, can impair the immune response against the pathogen and slow down regeneration. However, we focused on metabolic alterations, but are aware of the fact that psychological factors are very important and should be considered and incorporated. We discuss this topic in more detail according to the reviewer’s recommendation in the „Discussion“ and “Conclusions”.  Additionally, we analyzed the prevalence of psychosocial stress after 3 months in 97 of the 142 patients (11.34 % ) and whether there were gender differences:  A PCL-5 score >32 was reported more often by women: men: 10.5 % vs. women 12.5 %).

Conclusively, we also recommended not only to look at biomarkers, but rather use the biopsychosocial model and integrative health approaches for further studies.

Comments on the Quality of English Language

Minor editing of English language required. Several typos

We carefully revised the manuscript as recommended and corrected all typos we found.

Reviewer 2 Report

The manuscript describes the investigtion of possible amino acid metabolites and INFγ mediated pathways and their association with persistent symptoms after COVID-19. The work is actual and interesting and fits with the journal topics. I suggest the publication in the Journal after minor revision.

Questions and comments:

Explanation of some abbreviations is missing, e.g., ROS.

The paper by Lawer et al. (J. Proteome Res. 2021, 20, 2796-2811) should be discussed in the manuscript and added into the references.

Why only two amino acids were investigated? What about other proteinogenic amino acids?

The part Conclusions is missing.

The quality of the figures must be approved. The X and Y axis labels are hardly readable.

NA

Author Response

Dear Reviewers, 

First, we would like to thank you for taking the time to review our manuscript and for providing valuable suggestions on how to refine the manuscript for publication. Your effort and comments are very much appreciated! We revised our manuscript carefully according to the recommendations given, please find below our reply:

  • Explanation of some abbreviations is missing, e.g., ROS.

We added the explanations of the missing abbreviations.

  • Why were only two amino acids investigated? What about other proteinogenic amino acids? The paper by Lawler et al. (J. Proteome Res. 2021, 20, 2796-2811) should be discussed in the manuscript and added into the references.

Unfortunately, we only had little sample volume available, and thus could not measure other proteinogenic amino acids. However, such analyses are planned for further studies investigating metabolic alterations in patients with Long COVID, as this study and also the study by Lawler and coworkers indicate that COVID leads to impaired amino acid metabolism. We cited the paper by Lawler and coworkers and discussed it.

(See also paper by Lawler NG, Gray N, Kimhofer T, et al.; doi:10.1021/acs.jproteome.1c00052)

  • The part Conclusions is missing.

We added a paragraph with conclusions to the manuscript for further clarity.

  • The quality of the figures must be approved. The X and Y axis labels are hardly readable.

We revised and edited the figures for better comprehension and legibility.

Round 2

Reviewer 1 Report

The manuscript is much improved, however there are still some points to consider:

The Abstract still is not straight forward enough with regard to the design of the paper. You say that inflammatory parameters and symptoms are determined cross-sectionally. This is ok, however, then you mention a longitudinal approach (“after 60 days”). Maybe, you first start with what you write in the Participants and Design section of the paper: “Patients who had had acute, PCR-confirmed COVID-19 infection about 60 days earlier, were recruited within the scope of the prospective CovILDstudy that was …” and then move on to your cross-sectional approach.

The issue with the self-constructed questionnaires is now ok for me, however, please put a sentence in the limitation section that the questionnaires applied in this study are self-constructed and lack the usual quality criteria (objectivity, reliability, validity).

My biggest problem with the paper is still not satisfactorily solved. Still, you seem not to be able to discriminate those patients who have chronic immune-related issues before their acute COVID from those who did not have any immune-related issues before their acute COVID. Thus, your study cannot tell whether the “post-COVID” symptoms you correlated with inflammation cross-sectionally are attributable to COVID or (/and) to the underlying disease (which of course might have been responsible for the acute COVID disease as well). If this is true, you should at least discuss that and put it in the limitation section of the paper as well. You mentioned the problem in the last paragraph of the discussion. However, the point is not clear enough and some information should put in other sections of the paper, i.e. “As blood specimens from acute illness were only available from less than a third of patients,(Gietl et al [13]), we only performed a cross-sectional analysis of the data of 142 patients.”

English quality is quite good.

Author Response

Dear reviewer,

We revised our manuscript carefully according to your recommendations given, please find below our reply:

The Abstract still is not straight forward enough with regard to the design of the paper. You say that inflammatory parameters and symptoms are determined cross-sectionally. This is ok, however, then you mention a longitudinal approach (“after 60 days”). Maybe, you first start with what you write in the Participants and Design section of the paper: “Patients who had had acute, PCR-confirmed COVID-19 infection about 60 days earlier, were recruited within the scope of the prospective CovILDstudy that was …” and then move on to your cross-sectional approach.

We revised the abstract and added the recommended information for more clarity.

The issue with the self-constructed questionnaires is now ok for me, however, please put a sentence in the limitation section that the questionnaires applied in this study are self-constructed and lack the usual quality criteria (objectivity, reliability, validity).

The issue is addressed in the limitations of the study and explained in more detail.

My biggest problem with the paper is still not satisfactorily solved. Still, you seem not to be able to discriminate those patients who have chronic immune-related issues before their acute COVID from those who did not have any immune-related issues before their acute COVID. Thus, your study cannot tell whether the “post-COVID” symptoms you correlated with inflammation cross-sectionally are attributable to COVID or (/and) to the underlying disease (which of course might have been responsible for the acute COVID disease as well). If this is true, you should at least discuss that and put it in the limitation section of the paper as well. You mentioned the problem in the last paragraph of the discussion. However, the point is not clear enough and some information should put in other sections of the paper, i.e. “As blood specimens from acute illness were only available from less than a third of patients (Gietl et al [13]), we only performed a cross-sectional analysis of the data of 142 patients.”

As recommended we pointed out this limitation of this study.

Furthermore we performed further analyses investigating, whether patients without any comorbidities before Covid 19 infection differed from patients without comorbidities. Neopterin concentrations were lower in patients without comorbidities (n=31; median 8.2 nM vs. 9.7 nM), all other investigated lab parameters did not differ.

The number of comorbidities was not correlated with the investigated lab parameters, neither in the whole cohort nor in women or men. Higher neopterin levels were only found in men with co-morbidities (n=17), in women with comorbidities (n=14) neopterin levels were similar to patients without co-morbidities.

Patients with cardiovascular comorbidities (i.e. at least one of the following: obesity, diabetes, coronary artery disease/stroke, hypertension or hyperlipidemia; n= 58) had higher neopterin values (median 10.3 vs. 9.45 nM) and lower nitrite levels (median 18.5 vs. 26.7 µM, but higher Phe/Tyr (median 1.19 vs. 1.12).

We added this information to the “Results”.